

# How to optimize the absorption of two entangled photons

**Edoardo G. Carnio**[1,2,*], **Andreas Buchleitner**[1,2] **and Frank Schlawin**[3,4,5,6,†]

**1** Physikalisches Institut, Albert-Ludwigs-Universität Freiburg,
Hermann-Herder-Straße 3, D-79104 Freiburg, Germany
**2** EUCOR Centre for Quantum Science and Quantum Computing,
Albert-Ludwigs-Universität Freiburg, Hermann-Herder-Straße 3, D-79104 Freiburg, Germany
**3** Clarendon Laboratory, University of Oxford, Parks Road, Oxford OX1 3PU, United Kingdom
**4** The Hamburg Centre for Ultrafast Imaging,
Luruper Chaussee 149, D-22761 Hamburg, Germany
**5** Max Planck Institute for the Structure and Dynamics of Matter,
Luruper Chaussee 149, D-22761 Hamburg, Germany
**6** Universität Hamburg, Luruper Chaussee 149, D-22761 Hamburg, Germany

⋆ edoardo.carnio@physik.uni-freiburg.de, † frank.schlawin@mpsd.mpg.de

## Abstract

We investigate how entanglement can enhance two-photon absorption in a three-level system. First, we employ the Schmidt decomposition to determine the entanglement properties of the optimal two-photon state to drive such a transition, and the maximum enhancement which can be achieved in comparison to the optimal classical pulse. We then adapt the optimization problem to realistic experimental constraints, where photon pairs from a down-conversion source are manipulated by local operations such as spatial light modulators. We derive optimal pulse shaping functions to enhance the absorption efficiency, and compare the maximal enhancement achievable by entanglement to the yield of optimally shaped, separable pulses.

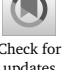

# 1   Introduction

Coherent control generally refers to the manipulation of quantum dynamical processes by suitably shaped control fields or interactions [1–3]. It has found widespread application in the control of chemical reactions [4–9], in spectroscopy [10, 11], in laser cooling [12–14] or even in quantum information and computing [15, 16]. In these well-established scenarios, the control is built on the manipulation of interfering pathways to maximize a desired outcome or speed up a process using classical laser pulses [17]. Optimal control theory then consists in finding the optimal laser pulse shapes and sequences, and the quantum character of light can be safely neglected.

In recent years, the possible use of *quantum* light sources for quantum-enhanced applications in microscopy or spectroscopy has gathered a lot of attention [18–23]. Of particular interest is the use of entangled photon pairs for applications which involve two-photon transitions. Such pairs can induce two-photon transitions more efficiently than laser pulses, and promise the use at low photon flux, thus preventing damage in photosensitive samples [24–27]. In addition, quantum correlations may help to further manipulate optical signals [28–30]. Yet, a key problem in the practical application, currently, is the low absorption cross section of many samples [31–34], unless the two-photon transition can proceed through near-resonant intermediate states [35]. In this case [36], the shaping of entangled photonic wave functions [37–39] can enhance the absorption probability. This strategy was explored in a number of recent theoretical papers, where the crucial role of quantum correlations between different travelling modes [40–42], or of the quantum statistics of a cavity mode [43], was highlighted. In contrast to classical control described by optimal control theory, the light fields have to be treated quantum mechanically, and, due to the small photon number per mode, perturbation theory can be employed.

Here we present a detailed study of how entanglement shared between two photons can enhance the probability to induce a two-photon excitation, in a sample which we describe by a simple three-level toy model, with finite excited state lifetimes, as depicted in Fig. 1(a). We carefully examine the optimal pulse shapes, as well as the relation between quantum correlations and the achievable enhancement. The insight obtained from this basic three-level

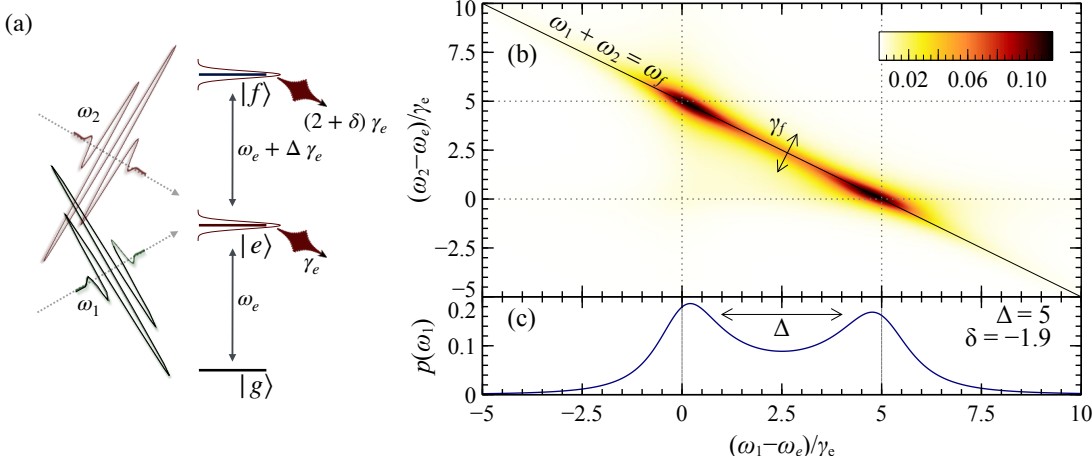

Figure 1: (a) A two-photon transition from ground state $|g\rangle$ to target state $|f\rangle$, via intermediate state $|e\rangle$, is driven by two incoming, near-resonant single-photon pulses with carrier frequencies $\omega_1$ and $\omega_2$, respectively. The excitation efficiency is subject to finite decay rates of $|e\rangle$ and $|f\rangle$, the characteristic scale $\omega_e$ of the level spacing, the detuning $\Delta$ of the individual nearest neighbour level spacings from degeneracy (in units of the intermediate state's lifetime $\gamma_e$), the deviation of the excited state decay rates' ratio from two (again in units of $\gamma_e$), see (17), and, here of central interest, the frequency correlations (see panel (b)) inscribed into the incoming pulses. (b) Density $|T_t(\omega_1, \omega_2)|^2 \propto |\Phi(\omega_1, \omega_2)|^2$ of the optimal two-photon wave function (32) in the space of rescaled frequencies $(\omega_j - \omega_e)/\gamma_e$, and parametrized by detuning $\Delta = 5$ and deviation $\delta = -1.9$ as specified in panel (a) and (17). (c) Single-photon density (34) derived from (32) upon tracing over the frequency of the partner photon in $|\Phi(\omega_1, \omega_2)|^2$, for the same parameter values as in panel (b). The double-peaked distributions in (b,c) feature maxima in the vicinity of the first ($|g\rangle \to |e\rangle$) and second ($|e\rangle \to |f\rangle$) bare transition's (see panel (a)) rescaled frequencies 0 and $\Delta$, respectively, with distinct widths determined by $\Delta$ and $\delta$, see (33,34). The widths of the local maxima give a qualitative impression of the nonclassical frequency correlation between both incoming photons.

unit applies also to two-photon transitions in more complex molecular spectra, where the interference between different excitation pathways has to be taken into account [44]. In a second part, we then demonstrate how this formalism can be applied to derive optimal states under realistic experimental conditions. In particular, we ask how a given entangled two-photon state can be optimized by only local operations on the individual photons, in order to induce the desired two-photon transition most efficiently. We find that, depending on the initial two-photon state and the sample, substantial enhancements of the absorption probability can be achieved.

The paper is organized as follows: in Sec. 2, we formulate the coherent control problem. We derive and solve functionals for the optimal quantum states of light to excite a simple three-level quantum system. In Sec. 3, we describe properties of ideal pulses, and analyze the relation between the entanglement they encode and the possible quantum advantage they offer (over optimal *separable* quantum states of light). Subsequently, in Sec. 4, we turn to the question of how entangled photons can be optimized for two-photon absorption (TPA) in experimentally realistic scenarios, before we conclude in Sec. 5.

## 2 Theoretical framework

### 2.1 The Hamiltonian

We consider the interaction between two (propagating) pulses of the quantized radiation field (the "field" degrees of freedom) and three electronic energy levels of an atom or molecule (the "matter" degrees of freedom). These systems are modelled, respectively, by Hamiltonians $H_f$ and $H_m$, and are coupled by an interaction term $W$. We describe all of these terms in the following paragraphs.

The incoming light pulses impinge on the three-level target located at the origin of our reference frame. Each field is quantized within a (cylindrical) quantization volume, of cross section $A$, along a distinct propagation direction, giving rise to modes labelled by a one-dimensional continuous variable, either the wave vector $k$ or the frequency $\omega$ [45]. Choosing the latter, for each beam $j$ we obtain annihilation $a_j(\omega)$ and creation $a_j^\dagger(\omega)$ operators satisfying the commutation relation $[a_j(\omega), a_l^\dagger(\omega')] = \delta_{jl}\delta(\omega - \omega')$. In this framework the photon number operator for beam $j$ reads $n_j = \int_0^\infty d\omega\, a_j^\dagger(\omega)a_j(\omega)$, while the total Hamiltonian for both fields is $H_f = \sum_j \int_0^\infty d\omega\, \hbar\omega a_j^\dagger(\omega)a_j(\omega)$. At the origin, where it interacts with the sample, the positive-frequency part of the electric field operator for the Hilbert space of photon $j$ (in the interaction picture starting at $t_0$ with respect to $H_f$) reads

$$E_j^+(t) = i \int_0^\infty d\omega \left(\frac{\hbar\omega}{4\pi\epsilon_0 cA}\right)^{1/2} a_j(\omega)e^{-i\omega(t-t_0)}, \tag{1}$$

where $\epsilon_0$ is the dielectric constant and $c$ the vacuum speed of light. We assume the fields have parallel polarization, but are distinguished by their propagation directions.

In the following we consider incoming photon pulses of bandwidth $\Delta\omega$ much smaller than their central frequency $\omega_0$, i.e. $\Delta\omega \ll \omega_0$. We can then employ the *narrow bandwidth approximation*, where we extend the range of all above frequency integrals to $(-\infty, \infty)$, substitute $\omega$ with $\omega_0$ within the square root in the integrand of (1), pull the resulting constant factor out of the integral, and define the Fourier-transformed annihilation operator

$$a_j(t) = \frac{1}{\sqrt{2\pi}} \int_{-\infty}^\infty d\omega\, a_j(\omega)e^{-i\omega(t-t_0)}. \tag{2}$$

Together with its adjoint $a_j^\dagger(t)$, it satisfies the commutation relation $[a_j(t), a_l^\dagger(t')] = \delta_{jl}\delta(t-t')$. The electric field operator (1) for beam $j$ is finally re-expressed as

$$E_j^+(t) = i\mathcal{E}_0 a_j(t), \text{ with } \mathcal{E}_0 = \sqrt{\hbar\omega_0/(2\epsilon_0 cA)}, \tag{3}$$

so that the total (positive-frequency) electric field seen by the atomic target reads (we neglect any geometry dependence in the coupling factors [23])

$$E^+(t) = E_1^+(t) + E_2^+(t) = i\mathcal{E}_0 [a_1(t) + a_2(t)]. \tag{4}$$

As shown in Fig. 1(a), the three non-degenerate electronic energy eigenstates of our target are $|g\rangle$, $|e\rangle$, and $|f\rangle$, with increasing energy. We define the origin of the energy axis to coincide with the energy of $|g\rangle$, hence the excited state energies are $\hbar\omega_e$ and $\hbar\omega_f$, respectively. With these labels, the matter Hamiltonian is $H_m = \hbar\omega_e |e\rangle\langle e| + \hbar\omega_f |f\rangle\langle f|$.

The light-matter coupling is mediated by an electric dipole Hamiltonian, switched on at $t_0$, which – in the interaction picture with respect to $H_f + H_m$, and for near-resonant perturbative driving of the atomic transitions – reads

$$W_I(t) = -V(t)E^-(t) - V^\dagger(t)E^+(t), \tag{5}$$

with $E^-$ the adjoint of $E^+$. For the specific level structure here considered (including the assumption that the atomic eigenstates have well-defined parity), the dipole operator, along the fields' polarization, has the explicit form

$$V(t) = \mu_{ge} e^{-i\omega_e(t-t_0)} |g\rangle \langle e| + \mu_{ef} e^{-i(\omega_f - \omega_e)(t-t_0)} |e\rangle \langle f|, \tag{6}$$

where the dipole matrix transition elements $\mu_{ge}$ and $\mu_{ef}$ between $|g\rangle$ and $|e\rangle$, and between $|e\rangle$ and $|f\rangle$, respectively, can be chosen real-valued.

## 2.2 Two-photon absorption amplitude

Our objective is to identify the optimal two-photon field state $|\Phi\rangle$ that drives the matter degrees of freedom from its initial (at time $t_0$) state $|g\rangle$ into the target state $|f\rangle$ (at time $t$), by TPA via $|e\rangle$. This is tantamount of maximizing the transition probability

$$p_f(t) = |\langle f(t), 0|U_I(t)|g, \Phi\rangle|^2, \tag{7}$$

where $|0\rangle$ indicates the vacuum state of both injected fields, $|f(t)\rangle = e^{i\omega_f(t-t_0)}|f\rangle$ exhibits the explicit time dependence of the interaction picture with respect to $H_m$ (whereas $|g\rangle$, at energy zero, remains unaffected), and $U_I(t)$ is the time evolution operator in the interaction picture of $H_f + H_m$, given by the Dyson series

$$U_I(t) = \mathbb{I} + \sum_{n=1}^{\infty} \left(\frac{1}{i\hbar}\right)^n \int_{t_0}^t d\tau_n \int_{t_0}^{\tau_n} d\tau_{n-1} \ldots \int_{t_0}^{\tau_2} d\tau_1 W_I(\tau_n) \ldots W_I(\tau_1). \tag{8}$$

Since $|\Phi\rangle$ is to be optimized in (7), while the initial atomic and field, as well as the final atomic state are fixed, we can extract the transition amplitude operator

$$T_{fg}(t) = \langle f(t)|U_I(t)|g\rangle \tag{9}$$

acting solely on the field degrees of freedom.[1]

## 2.3 Matter response function

Given the faint incoming field implied by (7) upon fixing the impinging two-photon state $|\Phi\rangle$, we can expand $U_I(t)$ in powers of $W_I$, such that the leading perturbative contribution to the desired two-photon transition reads, with (5,6,9),

$$T_{fg}(t) = \left(\frac{1}{i\hbar}\right)^2 \int_{t_0}^t d\tau_2 \int_{t_0}^{\tau_2} d\tau_1 \langle f(t)|W_I(\tau_2)W_I(\tau_1)|g\rangle. \tag{10}$$

Using (4) and (5), the integrand becomes

$$\langle f(t)|W_I(\tau_2)W_I(\tau_1)|g\rangle = -\mathcal{E}_0^2 \mu_{ge}\mu_{ef} e^{-i\omega_f t} e^{i(\omega_f-\omega_e)\tau_2} e^{i\omega_e\tau_1} [a_1(\tau_1)a_2(\tau_2) + a_1(\tau_2)a_2(\tau_1)], \tag{11}$$

where we dropped the terms $a_i(\tau_1)a_i(\tau_2)$ which result from $E^+(\tau_2)E^+(\tau_1)$ by (4), since the sought-after pulse $|\Phi\rangle$ has only one photon in each mode. Equation (2) and an exchange of the frequency and time integrations gives

$$T_{fg}(t) = \int_{-\infty}^{\infty} d\omega_1 \int_{-\infty}^{\infty} d\omega_2 \, T_{t;t_0}(\omega_1, \omega_2) a_2(\omega_2) a_1(\omega_1), \tag{12}$$

---

[1]Note that (7,9) imply a strictly unitary dynamics, and that rigorous account of incoherent processes beyond the purely phenomenological level adopted below would require a treatment in terms of the matter-field density matrix [46].

where $T_{t;t_0}(\omega_1, \omega_2)$ is the *matter response function* under weak field driving and for a finite interaction time starting at $t_0$, as made explicit by the index. Note that $T_{t;t_0}(\omega_1, \omega_2)$ must also encode the symmetry of (11) under exchange of the photon from the first or the second pulse being absorbed first. This will become explicit in (14).

To compactify the explicit expressions for $T_{t;t_0}(\omega_1, \omega_2)$, we introduce the line shape functions

$$\mathcal{L}_s(\omega) = \frac{i\mathcal{E}_0}{\sqrt{2\pi}\hbar} \frac{\mu_{s-1\,s}}{\omega - \omega_s + i\gamma_s}, \tag{13}$$

where the dipole matrix element $\mu_{s-1\,s}$ connects the state $s$ to the next energetically lower lying state $s-1$ (e.g., $|e\rangle$ to $|g\rangle$), and $\gamma_s$ phenomenologically accounts for finite decay rates of $|f\rangle$ and $|e\rangle$ ($|g\rangle$, being the ground state, cannot decay). With (10,11,13) we can thus extract from (12)

$$
\begin{aligned}
T_{t;t_0}(\omega_1, \omega_2) &= \frac{\mathcal{E}_0^2}{2\pi\hbar^2} \mu_{ge}\mu_{ef} e^{-i(\omega_f - i\gamma_f)t} e^{i(\omega_1 + \omega_2)t_0} \\
&\times \int_{t_0}^{t} d\tau_2 \int_{t_0}^{\tau_2} d\tau_1\, e^{-i[\omega_2 - (\omega_f - \omega_e) + i(\gamma_f - \gamma_e)]\tau_2} e^{-i(\omega_1 - \omega_e + i\gamma_e)\tau_1} + (\omega_1 \leftrightarrow \omega_2) \\
&= \mathcal{L}_e(\omega_1) \left\{ \left[ e^{-i(\omega_1 + \omega_2)(t-t_0)} - e^{-i(\omega_f - i\gamma_f)(t-t_0)} \right] \mathcal{L}_f(\omega_1 + \omega_2) \right. \\
&\left. - \left[ e^{-i(\omega_2 + \omega_e - i\gamma_e)(t-t_0)} - e^{-i(\omega_f - i\gamma_f)(t-t_0)} \right] \mathcal{L}_f(\omega_2 + \omega_e) \right\} + (\omega_1 \leftrightarrow \omega_2),
\end{aligned}
\tag{14}
$$

where $(\omega_1 \leftrightarrow \omega_2)$ in the last line indicates identical terms with the frequencies $\omega_1$ and $\omega_2$ of the two photons exchanged.

## 2.4 Infinitely extended pulses

Equation (14) describes the matter response at time $t$ to a pulse switched on at $t_0$, and thus vanishes for $t = t_0$. In the following, we will consider a scenario where the light-matter interaction is always switched on, while the case of finite $t - t_0$ will be discussed elsewhere. We therefore take the limit $t - t_0 \to \infty$, and, as a consequence, the real exponential factors in (14) vanish due to the excited state lifetimes, and we obtain the simpler expression [40]

$$T_t(\omega_1, \omega_2) = e^{-i(\omega_1 + \omega_2)(t-t_0)} [\mathcal{L}_e(\omega_1) + \mathcal{L}_e(\omega_2)] \mathcal{L}_f(\omega_1 + \omega_2), \tag{15}$$

which we will focus on hereafter. The remaining, global phase factor expresses the time dependences of $a_{1,2}(t)$ from (2) in the interaction picture of $H_f$; it therefore carries no physical significance – it drops out in the calculation of the probability (7) – and can be ignored in all our subsequent calculations.

Let us also observe that (15) can be easily adapted to a more intricate level structure of the target, with several intermediate states, as presented in [40]. All is needed is a $\sum_e$ in (6), which, by linearity, can be carried directly throughout all derivations above, to obtain:

$$T_t(\omega_1, \omega_2) = e^{-i(\omega_1 + \omega_2)(t-t_0)} \sum_e [\mathcal{L}_e(\omega_1) + \mathcal{L}_e(\omega_2)] \mathcal{L}_f(\omega_1 + \omega_2). \tag{16}$$

The structure of the matter response function in (15) directly reflects the absorption process in the matter: while the term $\mathcal{L}_e(\omega_1) + \mathcal{L}_e(\omega_2)$ describes the transition $|g\rangle \to |e\rangle$ induced by either one of the two photons, the term $\mathcal{L}_f(\omega_1 + \omega_2)$ correlates the two photons' frequencies by requiring that their sum be resonant with the two-photon transition $|g\rangle \to |f\rangle$. For this reason, the analysis presented in the next section rather depends on the detuning between the constituent one photon transitions of the two photon process under study. To quantify the departure from the spectrum of two non-interacting two-level systems (where $\omega_f = 2\omega_e$ and

$\gamma_f = 2\gamma_e$), which can be independently driven by one single-photon pulse each, we introduce the (dimensionless) detuning $\Delta$ and the deviation $\delta$ as

$$\Delta = \left(\omega_f - 2\omega_e\right)/\gamma_e,$$
$$\delta = \gamma_f/\gamma_e - 2,$$

(17)

as indicated in Fig. 1(a).

## 2.5 Connection to fluorescence measurements

Here we briefly comment on the experimental signatures of our maximization procedure. We envision that, after excitation of $|f\rangle$ by the two photons, the state decays by emitting a photon, which is measured in a photodetector. A similar experiment was carried out in hot rubidium vapour in [25], where the intermediate state was, however, off-resonant. To describe this experiment, a very similar derivation to ours was presented in [47] (compare, e.g., Eq. (31) there to Eq. (7) of the present manuscript). As pointed out there, the fluorescence signal induced by two-photon absorption of entangled photons is directly proportional to the population of $|f\rangle$ after the pulses have passed through the sample (and after a suitable ensemble average over many pulses or time). This condition is satisfied in the limit $t - t_0 \to \infty$ of Sec. 2.4, and, consequently, maximizing the population is directly equivalent to maximizing the fluorescence rate induced by a flux of entangled photon pairs, and we may write the fluorescence signal as [18]

$$S_{\text{fluorescence}} \propto \langle p_f(t)\rangle_{\text{ensemble}} \propto \int \mathrm{d}t\, p_f(t).$$

(18)

## 2.6 On notation

Let us conclude this theoretical introduction by clarifying, in one single place, the notation that we use throughout the next sections.

We deal with one-photon and two-photon states: the former are indicated with small, the latter with capital Greek letters. Consider a one-photon state, for a beam whose index we momentarily ignore. To refer to its Hilbert space element we use the common ket notation: $|\psi\rangle$. The *frequency representation* $\psi(\omega)$, or *wave function*, of the one-photon state $|\psi\rangle$ is its amplitude in the mode $\omega$:

$$\psi(\omega) = \langle\omega|\psi\rangle \iff |\psi\rangle = \int \mathrm{d}\omega\, \psi(\omega)a^\dagger(\omega)|0\rangle,$$

(19)

where we introduced the continuous-mode single-photon state [45]

$$|\omega\rangle = |1_\omega\rangle = a^\dagger(\omega)|0\rangle,$$

(20)

and the continuous-mode creation operator $a^\dagger(\omega)$ as presented in Sec. 2.1. For a two-photon state the notation and its interpretation are completely analogous, except for the indices of the two fields:

$$\Psi(\omega_1, \omega_2) = \langle\omega_1, \omega_2|\Psi\rangle \iff |\Psi\rangle = \iint \mathrm{d}\omega_1 \mathrm{d}\omega_2\, \Psi(\omega_1, \omega_2)a^\dagger(\omega_1)a^\dagger(\omega_2)|0\rangle.$$

(21)

Across Sec. 3 and Sec. 4 we switch between these two notations. The frequency representation eases the physical interpretation and makes some derivations mathematically more transparent. However, we switch to the Hilbert space when the derivations benefit from a more

compact notation. We spell out, to exemplify the change of notation, the functional derivative [48] of the overlap of two one-photon wave functions. In the frequency representation we write

$$\mathcal{F}[\phi, \psi^*] = \int d\omega\, \phi(\omega)\psi^*(\omega) \iff \frac{\delta\mathcal{F}}{\delta\psi^*} = \phi(\omega),$$ (22)

which is equivalent to writing, in the Hilbert space,

$$\mathcal{F}[|\phi\rangle, \langle\psi|] = \langle\psi|\phi\rangle \iff \frac{\delta\mathcal{F}}{\delta\langle\psi|} = |\phi\rangle.$$ (23)

## 3 Ideal pulses

### 3.1 Fluctuation-constrained optimization

Our goal is to maximize the population $p_f(t)$ in state $|f\rangle$ at the time $t$, by appropriate choice of the two-photon state $|\Phi\rangle$. The latter is determined [40] by the extrema of the functional

$$J[|\Phi\rangle] = p_f(t) - \lambda\left(\langle\Phi|n_1 n_2|\Phi\rangle - 1\right),$$ (24)

where the dependence of $p_f(t)$ on the state $|\Phi\rangle$ can be unveiled by inserting (9) into (7), to obtain

$$p_f(t) = |\langle 0|T_{fg}(t)|\Phi\rangle|^2 = \langle\Phi|T_{fg}^\dagger(t)|0\rangle\langle 0|T_{fg}(t)|\Phi\rangle.$$ (25)

The second term in (24) constrains the distribution of the injected two photons over the incoming fields via the Lagrange multiplier $\lambda$: out of all two-photon states, the expectation value $\langle n_1 n_2\rangle$ limits the search to those where each beam is populated by one photon.[2] Rather than by saturating the number of photons in either beam [19], then, we allow the maximization of $p_f(t)$ via quantum correlations [49], the potential of which we want to scrutinize here.

At an extremum of $J[|\Phi\rangle]$, its functional derivative, (22,23), must vanish,

$$\frac{\delta J}{\delta\langle\Phi|} = 0.$$ (26)

With (25) in (24), this requirement results in the eigenvalue problem

$$T_{fg}^\dagger(t)|0\rangle\langle 0|T_{fg}(t)|\Phi\rangle = \lambda|\Phi\rangle,$$ (27)

which, with the definition

$$|T\rangle = T_{fg}^\dagger(t)|0\rangle,$$ (28)

turns into

$$|T\rangle\langle T|\Phi\rangle = \lambda|\Phi\rangle.$$ (29)

In addition, requiring the variation of the functional (24) with respect to the Lagrange multiplier to vanish, i.e. $\delta J/\delta\lambda = 0$, enforces the normalization of the two-photon state, such that we arrive at

$$|\Phi\rangle = \mathcal{N}^{-1/2}T_{fg}^\dagger(t)|0\rangle$$ (30)

as the only solution.[3] The maximal population can thus be expressed directly in terms of $\mathcal{N}$, by (30) in (25), together with $\langle\Phi|\Phi\rangle = 1$ as imposed by the constraint in (24), or in frequency

---

[2]The expectation value is non-vanishing only for this configuration. Furthermore, (21) implies $\langle\Phi|n_1 n_2|\Phi\rangle = \langle\Phi|\Phi\rangle$, such that the Lagrange multiplier also enforces the normalization of the state.

[3]Since $|T\rangle\langle T|$ is a one-dimensional projector, it has only one non-trivial eigenstate.

space, with (12,13,15) in (30) in (25), to obtain an explicit expression for $\mathcal{N}$ in terms of the system parameters [40]:

$$p_f(t) = \mathcal{N} = \iint |T_t(\omega_1, \omega_2)|^2 \, d\omega_1 \, d\omega_2 = \frac{2\pi^2 (\mu_{ge}\mathcal{E}_0)^2 (\mu_{ef}\mathcal{E}_0)^2}{\hbar^4 \gamma_e \gamma_f}. \tag{31}$$

We remark here that the maximal population does *not* depend on time, yet it is attained at $t$ given the initial time $t_0$. This is due to the time dependence carried by the matter response function (15) in the interaction picture. As long as $t - t_0 \gg \gamma_e^{-1}$, the optimization problem will yield the optimal population (31) for arbitrary choice of $t_0$ and $t$.

## 3.2 Optimal two-photon states

To discuss the properties of the optimal two-photon state, we move to the frequency representation, following the prescriptions illustrated in Sec. 2.6. With (12) in (30), the optimal two-photon wave function (21) is given by

$$\Phi(\omega_1, \omega_2) = \langle \omega_1, \omega_2 | \Phi \rangle = \frac{T_t^*(\omega_1, \omega_2)}{\sqrt{\mathcal{N}}}, \tag{32}$$

which determines all the statistical properties of the two injected photons. Up to normalization, the optimal two-photon wave function in frequency space is the complex conjugate of the matter response function (15). This means that the state which maximizes the population in $|f\rangle$ at time $t$ is given by the time-reversed two-photon state emitted by the three-level system initially (at $t$) prepared in $|f\rangle$. This is the direct two-photon analogue of the well-known result of the optimal, "exponentially rising" single-photon state to excite a two-level atom [50,51], which is simply the time reversed version of the single photon wavepacket emitted by an excited two-level atom.

Figure 1(b) shows the properties of the optimal two-photon state as defined by (32), and parametrized by detuning $\Delta$ and deviation $\delta$. The concentration of $|T_t(\omega_1, \omega_2)|^2 \propto |\Phi(\omega_1, \omega_2)|^2$ around the anti-diagonal $\omega_1 + \omega_2 = \omega_f$ reflects the constraint, introduced by $\mathcal{L}_f(\omega_1 + \omega_2)$ in (15), and discussed in 2.4, that the total frequency of the injected photons be resonant with the two-photon transition $|g\rangle \rightarrow |f\rangle$. The frequency sum distribution is readily extracted from $|\Phi(\omega_1, \omega_2)|^2$, by changing variables to $\omega_\pm = \omega_1 \pm \omega_2$ and integrating over $\omega_-$:

$$p_{\text{sum}}(\omega_+) = \frac{1}{\pi} \frac{\gamma_f}{(\omega_+ - \omega_f)^2 + \gamma_f^2}, \tag{33}$$

i.e. a Lorentzian of width $\gamma_f = (2 + \delta)\gamma_e$, centred at $\omega_f = 2\omega_e + \Delta\gamma_e$.

In Fig. 1(c), instead, we plot the single-photon distribution, i.e. the marginal distribution of $|\Phi(\omega_1, \omega_2)|^2$ with respect to either frequency (here the first), since the optimal two-photon wave function (32) is symmetric:

$$p_1(\omega) = \frac{\gamma_e(\gamma_e + \gamma_f)(4\gamma_e + \gamma_f) + \gamma_e(\omega_f - 2\omega_e)^2 + \gamma_f(\omega - \omega_e)^2}{2\pi [(\omega - \omega_e)^2 + \gamma_e^2][(\omega - \omega_f + \omega_e)^2 + (\gamma_e + \gamma_f)^2]}, \tag{34}$$

where we can identify two distinct peaks near[4] the transitions $|g\rangle \rightarrow |e\rangle$ (frequency $\omega_e$ and line width $\gamma_e$) and $|e\rangle \rightarrow |f\rangle$ (frequency $\omega_f - \omega_e = \omega_e + \Delta\gamma_e$ and line width $\gamma_f + \gamma_e = (3 + \delta)\gamma_e$). Notice that, since $\Phi_t(\omega_1, \omega_2)$ is symmetric under exchange of variables, the single-photon frequency distribution is the same for both fields: each photon has the same probability to induce the transition to $|e\rangle$, and the second completes it to $|f\rangle$.

---

[4]The peaks do not exactly match the transition frequencies because $|\mathcal{L}_e(\omega_1) + \mathcal{L}_e(\omega_2)|^2 \neq |\mathcal{L}_e(\omega_1)|^2 + |\mathcal{L}_e(\omega_2)|^2$, and interference terms contribute to $p_1(\omega)$.

When $\delta \to -2$, that is in the limit of a final state with vanishing linewidth $\gamma_f \to 0$, $p_{\text{sum}}(\omega_+)$ tends to $\delta(\omega_+ - \omega_f)$. This describes the situation where the frequency of one "emitted" photon strictly determines that of the other, which means that each photon has the same probability of exciting either the $|g\rangle \to |e\rangle$ or the $|e\rangle \to |f\rangle$ transition. In this limit, $p_1(\omega)$ exhibits two peaks of equal (unit) width at $\omega_e$ and $\omega_f - \omega_e$, as discussed above.

### 3.3 Entanglement entropy

When $\Delta = \delta = 0$, i.e., by (17), $\omega_f = 2\omega_e$ and $\gamma_f = 2\gamma_e$, the optimal two-photon wave function given by (32) is separable: either photon independently excites one of two non-interacting two-level systems, see Sec. 2.4. Otherwise, the frequency degrees of freedom of the two optimally prepared incoming pulses are, in general, entangled [40], with non-trivial (i.e., $L > 1$) Schmidt decomposition into Schmidt modes $\varphi_k^*(\omega_1), \psi_k^*(\omega_2)$ and their non-increasingly ordered, non-negative Schmidt coefficients $r_k$,

$$\Phi(\omega_1, \omega_2) = \sum_{k=1}^{L} r_k \varphi_k^*(\omega_1) \psi_k^*(\omega_2), \tag{35}$$

where normalization via $\mathcal{N}$ in (32) implies $\sum_k r_k^2 = 1$ ($0 \le r_k \le 1$), and complex conjugation is inherited from that of $T_t(\omega_1, \omega_2)$, also in (32). The entanglement encoded in $\Phi(\omega_1, \omega_2)$ can be quantified by the entanglement entropy [49],

$$S = -\sum_{k=1}^{L} r_k^2 \log_2 r_k^2. \tag{36}$$

We now inspect how the degree of frequency entanglement of the incoming pulses correlates with enhanced absorption probabilities.

### 3.4 Quantum enhancement

By (31), the maximal population of the state $|f\rangle$ is $\mathcal{N}$. To capture the genuine advantage due to entanglement, an entangled fields' yield is to be compared with that of the optimal *separable* two-photon state, i.e. a two-photon state which does not feature entanglement between the two modes.[5] As shown in [40], the optimal separable state can be obtained with the same optimization procedure of Sec. 3.1, and it is simply given by the modes pertaining to the largest Schmidt coefficient $r_1$ in (36),

$$\Phi_{\text{sep}}(\omega_1, \omega_2) = \varphi_1^*(\omega_1) \psi_1^*(\omega_2). \tag{37}$$

Such an optimal classical state yields an excited state population $r_1^2 \mathcal{N} \le \mathcal{N}$, where equality is achieved only for $\Delta = \delta = 0$. The *quantum enhancement* of TPA achievable by an entangled two-photon state, through the quantum correlations between the incoming fields, is thus given by the ratio

$$E_q = r_1^{-2} \ge 1 \tag{38}$$

of those two excitation probabilities.

According to (36), strong entanglement $S$ requires $L \gg 1$ and thus small $r_1$, which implies strong quantum enhancement $E_q$, by (38). We test this mutual relationship in Fig. 2, where we evaluate (37) and (38) with the optimal two-photon wave function (32), for variable

---

[5]Note that, in comparing the efficiency of two-photon states only, we are not considering the impact of "photon number entanglement", i.e. the role of photon statistics.

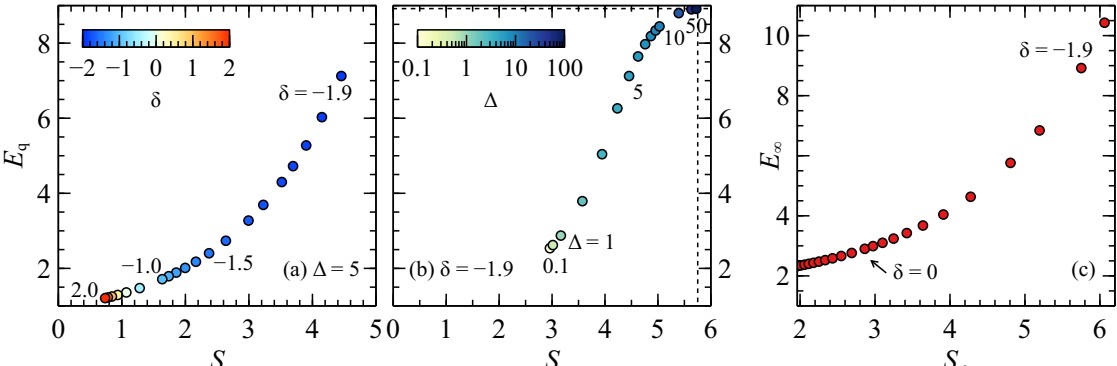

Figure 2: Entanglement entropy $S$ as defined in (36), and quantum enhancement $E_q$ given in (38), of the excited state population $p_f(t)$ (25) achieved by the optimal two photon state (32) for (a) fixed detuning $\Delta = 5$ with variable deviation $\delta \in (-2; 2]$ (note the colour code, and specific values indicated by labels), and (b) fixed deviation $\delta = -1.9$ with variable detuning $\Delta \in [0.1; 100]$ (colour coded and labeled). The dashed lines indicate the maximally achievable entanglement and quantum enhancement, respectively, in the limit $\Delta \to \infty$. (c) Maximally achievable enhancement $E_\infty$, see (43) and Sec. 3.5, and associated entanglement entropy $S_\infty$, given by (44), as a function of $\delta$, in the limit $\Delta \to \infty$ (achieved by $\gamma_e \to 0$).

deviation $\delta$ (panel (a)) and detuning $\Delta$ (panel (b)), respectively, as well as the maximally achievable values of entanglement and enhancement, $S_\infty$ and $E_\infty$, respectively, in the limit of very large $\Delta$ (panel (c)), which we discuss separately in Sec. 3.5.[6] In panels (a,b), we observe a monotonic increase of $E_q$ with $S$, and a plateau of the achievable entanglement and quantum enhancement emerges for large values of the detuning $\Delta$, which we address in the next subsection. The increase of $S$ with $\Delta$ and as $\delta \to -2$ can be attributed to a narrowing of the frequency-sum distribution (33), in unison with a broad single frequency distribution (34) (due to its composition by two distant peaks). This distribution signifies strong frequency anti-correlations, and indeed, since we are here considering pure quantum states of light, correspond to a strongly entangled wave function [19].

## 3.5 Maximal quantum enhancement

Let us now discuss the quantum enhancement $E_q$ in the limit $\Delta \gg 1$ (or, more physically, $\omega_f - 2\omega_e \gg \gamma_e$), which Fig. 2(b) suggests to be finite. As discussed in Sec. 2.4, the matter response function (15) is symmetric in the two photon frequencies, which implies that its Schmidt modes $\varphi_k, \psi_k$ in (35) are equal for each $k$, possibly up to a phase. For $\Delta \gg 1$, as in Fig. 3, said modes appear as orthogonal superpositions of non-overlapping, complex-valued line shapes $\alpha_k$ (centred at $\omega_e$) and $\beta_k$ (centred at $\omega_f - \omega_e = \omega_e + \Delta\gamma_e$). Moreover, mutually orthogonal linear combinations of the same modes, e.g. $\varphi_{1,2} = (\alpha_1 \pm \beta_1)/\sqrt{2}$ in Fig. 3, are associated with Schmidt coefficients $r_1 \approx r_2$, at least for finite $\Delta$.

Let us now understand the connection between the Schmidt modes $\varphi_k, \psi_k$ and the line shapes $\alpha_k, \beta_k$. The latter can be considered the Schmidt modes of a response function where the first photon is resonant with the $|g\rangle \to |e\rangle$ transition, while the second completes the

---

[6]All results here displayed are based on the Schmidt decomposition (35) of the two-photon wave function $\Phi(\omega_1, \omega_2)$ discretized in frequency space, i.e., we used a linear algebra package (Wolfram Mathematica) to calculate the singular values of a matrix. Frequencies were discretized on a grid of size $\pm 200\gamma_f$ and resolution $\gamma_e/5$. This choice of grid size and resolution was validated by a normalization of the discretized state which was systematically bounded from below by 0.99.

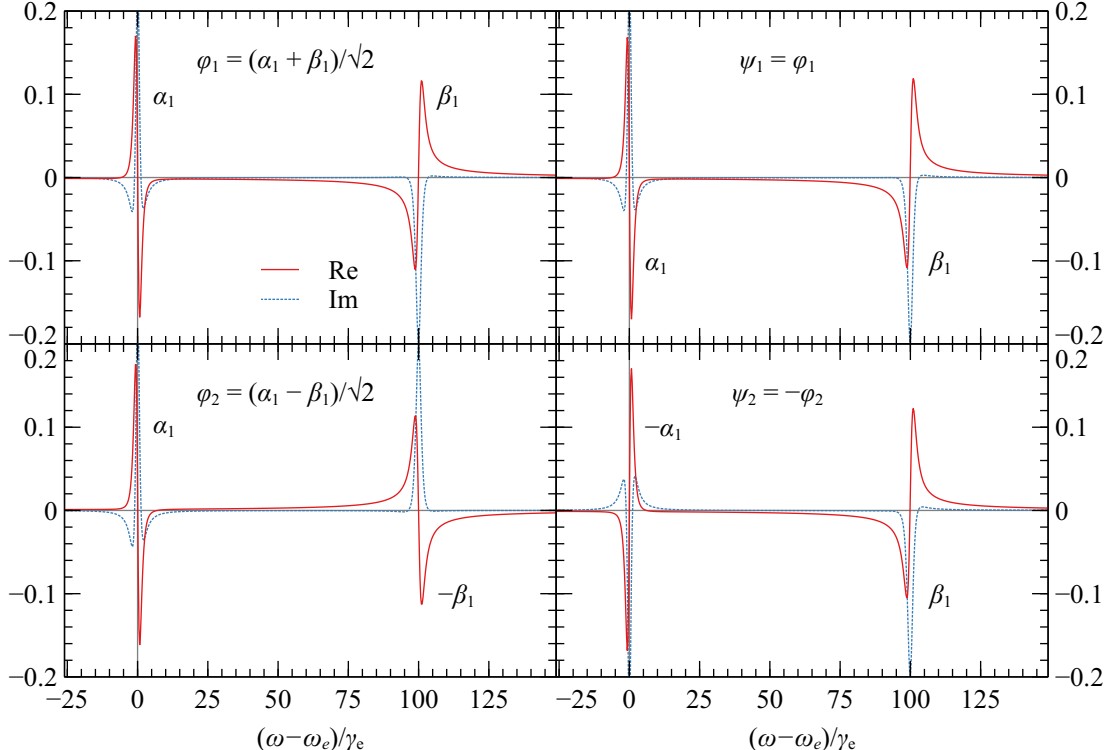

Figure 3: First two pairs $\varphi_k(\omega)$, $\psi_k(\omega)$ of Schmidt modes computed for a response function (15) with $\Delta = 100$ and $\delta = -1.5$, discretized on a grid $[-500; 500]$ with a resolution $(2+\delta)/2$. These pairs correspond to the Schmidt coefficients $r_1 = 0.272557$ and $r_2 = 0.272348$. Because of the large value of $\Delta$, the Schmidt modes are linear combinations of modes $\alpha_k$ and $\beta_k$ centred, respectively, at rescaled frequencies $(\omega - \omega_e)/\gamma_e$ 0 and $\Delta$.

two-photon transition by inducing $|e\rangle \to |f\rangle$. The construction of such a response function yields

$$Q_t(\omega_1, \omega_2) = e^{-i(\omega_1 + \omega_2)(t - t_0)} \mathcal{L}_e(\omega_1) \mathcal{L}_f(\omega_1 + \omega_2). \tag{39}$$

The analogy to (15) becomes clear by noticing that

$$T_t(\omega_1, \omega_2) = Q_t(\omega_1, \omega_2) + Q_t(\omega_2, \omega_1), \tag{40}$$

that is, $T_t$ is the symmetrized[7] version of $Q_t$. For visualization, $Q_t(\omega_1, \omega_2)$ and $Q_t(\omega_2, \omega_1)$ describe, respectively, the top-left and bottom-right peaks of $T_t(\omega_1, \omega_2)$ in Fig. 1(b).

Let us now write the Schmidt decomposition of the asymmetric response function (39) as

$$Q_t(\omega_1, \omega_2) = \sqrt{\frac{\mathcal{N}}{2}} \sum_{k=1}^{L'} s_k \alpha_k(\omega_1) \beta_k(\omega_2), \tag{41}$$

where the prefactor ensures $\sum_k s_k^2 = 1$. The Schmidt coefficients must be independent of $\Delta$, since this parameter translates the top-left peak in Fig. 1(b) without changing its anti-diagonal structure. This cannot be true for $T_t$ in (15), where a change in $\Delta$ implies both a vertical and a horizontal translation of, respectively, the top-left and bottom-right peaks. In this case, then, the structure on the anti-diagonal changes and so do the correlations between the two photon frequencies.

---

[7]It hence allows each photon to begin the two-photon transition, and the other to complete it.

Substituting (35) (on the left-hand side, via (32)) and (41) (on the right-hand side) in the identity (40) we obtain

$$\frac{T_t(\omega_1, \omega_2)}{\sqrt{\mathcal{N}}} = \sum_{k=1}^{L} r_k \varphi_k(\omega_1) \psi_k(\omega_2) = \sum_{l=1}^{L'} \frac{s_l}{\sqrt{2}} \left[\alpha_l(\omega_1)\beta_l(\omega_2) + \beta_l(\omega_1)\alpha_l(\omega_2)\right].$$

Going back to the example of Fig. 3, if we assume $r_1 = r_2$ and multiply out the combinations proposed for $\varphi_k(\omega_1)$ and $\psi_k(\omega_2)$, for $k = 1, 2$, we obtain precisely the products of $\alpha_1$ and $\beta_1$ on the right-hand side of (42). Notice that these combinations are not arbitrary: as discussed earlier, because $T_t$ is a symmetric complex function, the modes $\varphi_k$ and $\psi_k$ might differ by a phase that ensures the positivity of the singular values. All in all, we must satisfy $\varphi_1(\omega_1)\psi_1(\omega_2) + \varphi_2(\omega_1)\psi_2(\omega_2) = \alpha_1(\omega_1)\beta_1(\omega_2) + \beta_1(\omega_1)\alpha_1(\omega_2)$.

The right-hand side of (42) is a valid Schmidt decomposition *only* in the limit $\Delta \to \infty$, where the function bases $\{\alpha_k\}_k$ and $\{\beta_k\}_k$ are also mutually orthogonal,[8] since – to reconstruct the symmetry of $T_t$ – they both have to appear as Schmidt modes for each photon frequency. Under this condition, then, the Schmidt coefficients $r_k$ of the response function $T_t$ must come in pairs. In summary, in the limit $\Delta \to \infty$, (42) is a valid Schmidt decomposition and we can identify

$$r_{2k} = r_{2k-1} = \frac{s_k}{\sqrt{2}}, \quad \varphi_{2k-1} = \psi_{2k} = \alpha_k, \quad \varphi_{2k} = \psi_{2k-1} = \beta_k, \qquad k = 1, \ldots, L'. \tag{42}$$

The pairwise appearance of Schmidt coefficients $r_1 = s_1/\sqrt{2}$, by (42), implies that the enhancement $E_\infty$, when $\Delta \gg 1$, is twice the enhancement $E_a$ obtainable with the asymmetric matter response function (39):

$$E_\infty = 2E_a. \tag{43}$$

For the entropy, instead, we can write

$$S_\infty = -\sum_{k=1}^{L} r_k^2 \log_2 r_k^2 = -2\sum_{l=1}^{L'} \frac{s_l^2}{2} \log_2\left(\frac{s_l^2}{2}\right) = 1 + S_a, \tag{44}$$

where $S_a$ is the entanglement entropy of (41). For any value of $\delta$, then, the quantum enhancement induced by the optimal pulse (32), and the entanglement between the two photons' frequencies must be bounded, respectively, by $E_\infty$ and $S_\infty$, which are plotted in Fig. 2(c). For $\delta \to -2$ we observe the same steep increase in enhancement due to the "strict" correlation between the photon frequencies discussed in Secs. 3.2 and 3.4.

# 4 Realistic pulses

So far, our investigation targeted the optimal two-photon quantum state which can be constructed theoretically. However, the experimental manipulation of the two-photon state is subject to further constraints, and it is therefore necessary to reformulate the above theory for experimentally implementable transformations. This is the purpose of the present section. We consider an experiment in which a two-photon state $|\Sigma\rangle$ can be generated. The frequency components of this state will then be manipulated, e.g. using a spatial light modulator [37–39], to enhance the propensity of the state to excite a two-photon transition in the three-level sample it impinges on, as in Fig. 4(c). First we analyze a general class of transformations, and then consider two realistic examples (see figures 4(a) and (b)), where the two-photon state

---

[8]The overlap of these line shapes goes to zero in the limit $\Delta \to \infty$.

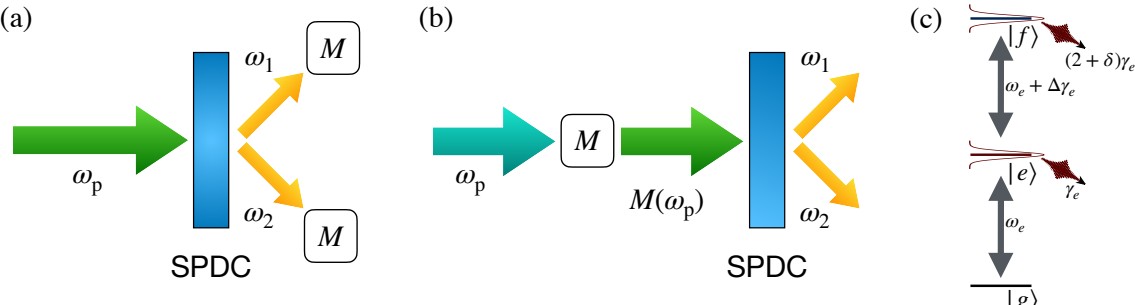

Figure 4: Exemplary experimental scenarios discussed in (a) Sec. 4.3 and (b) Sec. 4.4. In (a) two photons of frequencies $\omega_1$ and $\omega_2$ are produced via spontaneous parametric down-conversion (SPDC) from a pump photon of frequency $\omega_p$. Both photons are then frequency-modulated by the same pulse shaping operator $M$. In (b), instead, it is the pump photon that is shaped by an optimal operator $M$, before being split into two photons by SPDC. (c) recalls, for convenience, the parameters describing the matter degrees of freedom as in Fig. 1(a).

generated e.g. by spontaneous parametric down-conversion is spectrally shaped (a), or where the (classical) pump pulse that creates the entangled photon pair by down-conversion is transformed (b). As we will see, both situations can be analyzed in terms of unitary transformations acting on the two-photon state $|\Sigma\rangle$.

## 4.1 Optimization procedure

We want to find unitary operations, $M_1$ and $M_2$, which act on the Hilbert spaces of the first and of the second photon, respectively, and maximize the TPA probability induced by the manipulated state $M_1 M_2 |\Sigma\rangle$. If we continue to denote with $|\Phi\rangle$ the optimal state (30) analyzed in Sec. 3, we intuitively want $M_1 M_2 |\Sigma\rangle$ to most closely resemble $|\Phi\rangle$.

In the frequency representation of Sec. 2.6, we write a pulse shaping *function* as

$$\langle \omega_j | M_j | \nu_j \rangle = M_j(\nu_j, \omega_j). \tag{45}$$

In Sec. 4.3, the arguments in the above equation refers to the frequency components of the photon wavepacket in beam $j$. As we will see in Sec. 4.4, the same formalism can be applied to the shaping of the pump pulse in type-I parametric down-conversion, where, however, the frequency arguments represent the components of the sum frequency $\omega_1 + \omega_2$.

While in sections 4.2–4.4 we assume $M$ to be diagonal in frequency space, here we include operations that change the spectral shape of the photon wavepacket, as described, e.g., in Sec. V of [52]. We do, however, assume that the photon number is conserved, i.e. squeezing or displacement operations are not considered. As in Sec. 3, we want to maximize the final state population (7), but with the shaped two-photon state $M_1 M_2 |\Sigma\rangle$:

$$p_f(t) = |\langle 0 | T_{fg}(t) M_1 M_2 |\Sigma\rangle|^2 = \mathcal{N} |\langle \Phi | M_1 M_2 |\Sigma\rangle|^2. \tag{46}$$

The factor $\mathcal{N}$ stems from (30), and clarifies the meaning of this equation: if $\mathcal{N}$ is the maximal population achievable by using the optimal state $|\Phi\rangle$, then the population in the case of a shaped realistic state $M_1 M_2 |\Sigma\rangle$ is reduced by the overlap between these two initial states.

The new functional to optimize – instead of (24) – is then

$$J[M_1, M_2] = p_f(t) - \sum_{j=1}^{2} \left[ \langle \psi_j | M_j^\dagger M_j | \psi_j \rangle - \langle \psi_j | \psi_j \rangle \right]. \tag{47}$$

We remark here that (24) was maximized over the space of two-photon states, whereas now we are optimizing in the space of *operators* on the Hilbert spaces of the individual photons. In (47) we constrain the search to local operators that do not change the number of photons of undetermined, and potentially unnormalized, single-photon states $|\psi_j\rangle$ as per (19). The summation term in (47) therefore limits our search to unitary local operators:

$$M_j^\dagger M_j = \mathbb{I}. \tag{48}$$

The single-photon states $|\psi_j\rangle$ do not carry further relevance beyond imposing the constraint (48), and effectively play the same role of the Lagrange multiplier of (24).

Due to the higher dimensionality of the search space, we cannot find an explicit *general* expression of the optimal pulse shaping operators $M_j$. These are solutions of coupled integral equations (87), whose derivation we defer to Appendix A.

## 4.2 Diagonal pulse shaping operators

We now collect the theoretical underpinning of our subsequent discussion of examples in sections 4.3 and 4.4. We focus on *diagonal* pulse shaping operators, i.e. of the form

$$\langle \omega_j | M_j | \nu_j \rangle = M_j(\omega_j)\delta(\omega_j - \nu_j). \tag{49}$$

This means that we restrict ourselves to linear optical elements that do not change the photon energies, such as the aforementioned manipulation by a spatial light modulator. Substituting (49) in the derivation of Appendix A, we obtain, in place of (87),

$$M_j(\omega_j) = \frac{\sqrt{\mathcal{N}}\mathcal{A}[M_1,M_2]}{|\psi_j(\omega_j)|^2} \int \mathcal{W}_t^*(\omega_j,\omega_k)M_k^*(\omega_k)\,\mathrm{d}\omega_k. \tag{50}$$

The functional $\mathcal{A}[M_1,M_2]$ is defined in (81), and we have also introduced the *effective* response function $\mathcal{W}_t$ as the product of the matter response function (15, 32) with the (frequency representation of the) input state $|\Sigma\rangle$:

$$\mathcal{W}_t(\omega_1,\omega_2) = \Sigma(\omega_1,\omega_2)T_t(\omega_1,\omega_2) = \sqrt{\mathcal{N}}\Sigma(\omega_1,\omega_2)\Phi^*(\omega_1,\omega_2). \tag{51}$$

Using the effective response function, the final state population (46) achievable via the diagonal pulse shaping functions $M_1$ and $M_2$ of (49) reads

$$p_f(t) = \left|\iint \mathcal{W}_t(\omega_1,\omega_2)M_1(\omega_1)M_2(\omega_2)\,\mathrm{d}\omega_1\,\mathrm{d}\omega_2\right|^2. \tag{52}$$

Given the fact that $M_j$ is a unitary diagonal operator, see (48), we can write

$$M_j(\omega_j) = \mathrm{e}^{-iF_j(\omega_j)}, \tag{53}$$

with $F_j(\omega_j)$ a real function. Substitution of this ansatz in (50) yields

$$|\psi_j(\omega_j)|^2\mathrm{e}^{-iF_j(\omega_j)} = \sqrt{\mathcal{N}}\iint \mathcal{W}_t(\omega_1,\omega_2)\mathrm{e}^{-i[F_1(\omega_1)+F_2(\omega_2)]}\mathrm{d}\omega_1\mathrm{d}\omega_2\int \mathcal{W}_t^*(\omega_j,\omega_k)\mathrm{e}^{iF_k(\omega_k)}\mathrm{d}\omega_k. \tag{54}$$

If we also cast the effective response function in polar form,

$$\mathcal{W}_t(\omega_1,\omega_2) = |\mathcal{W}_t(\omega_1,\omega_2)|\mathrm{e}^{i\mathcal{S}(\omega_1,\omega_2)}, \tag{55}$$

the identity (54) is satisfied by functions $F_1$ and $F_2$ such that

$$F_1(\omega_1) + F_2(\omega_2) = \mathcal{S}(\omega_1,\omega_2), \tag{56}$$

and by a Lagrange multiplier $\psi_j$ with

$$|\psi_j(\omega_j)|^2 = \sqrt{\mathcal{N}} \iint |\mathcal{W}_t(\omega_1, \omega_2)| d\omega_1 d\omega_2 \int |\mathcal{W}_t(\omega_j, \omega_k)| d\omega_k. \tag{57}$$

Substituting (53) and (55) in (52), and assuming (56) is satisfied, we find the final state population achievable upon manipulating ("s" for "shaped") the input state $|\Sigma\rangle$:

$$p_f^{(s)}(t) = \left( \iint |\mathcal{W}_t(\omega_1, \omega_2)| d\omega_1 d\omega_2 \right)^2. \tag{58}$$

On the other hand, in the absence of shaping operators, that is with $M_1 = M_2 \equiv 1$, (52) yields

$$p_f^{(u)}(t) = \left| \iint \mathcal{W}_t(\omega_1, \omega_2) d\omega_1 d\omega_2 \right|^2. \tag{59}$$

This quantity is the "unshaped" or "unoptimized" population achievable when we do not manipulate the incoming realistic pulse. Its value is determined by the sign taken by the effective response function $\mathcal{W}_t$, i.e. by the phase difference between the optimal $|\Phi\rangle$ and the realistic state $|\Sigma\rangle$. These phase differences are precisely what the optimal[9] pulse shaping functions compensate, by transforming $\mathcal{W}_t(\omega_1, \omega_2)$ into $|\mathcal{W}_t(\omega_1, \omega_2)|$ via (53) with (56). To capture the enhancement obtained by pulse shaping we therefore introduce the *optimization ratio* as the ratio between optimized and unoptimized populations:

$$E_{\text{opt}} = \frac{p_f^{(s)}(t)}{p_f^{(u)}(t)}. \tag{61}$$

To conclude, in the following examples we will identify the effective response function $\mathcal{W}_t$. If we can determine, via its phase, the argument $F_j(\omega_j)$ of the pulse shaping operators, such that (56) is satisfied, then the optimal final state population is directly given by (58).

## 4.3 Example I: shaping down-converted photons

The first example we consider is depicted in Fig. 4(a): a pump photon of frequency $\omega_p$ is split via SPDC into two photons of frequencies $\omega_1$ and $\omega_2$, which are modulated by the same pulse shaping operator $M$. This setup describes the experiment carried out by the Silberberg group in 2005 [25]. We consider a frequency representation (21) of $|\Sigma\rangle$ given by

$$\Sigma(\omega_1, \omega_2) = \delta(\omega_1 + \omega_2 - \omega_p) G(\omega_1 - \omega_p/2), \tag{62}$$

where $\omega_p$ is the frequency of the pump photon creating the entangled pair, which we assume to be on resonance with the two-photon transition ($\omega_p = \omega_f$) in the matter, see Fig. 4(c). The delta function stems from a narrowband continuous-wave (cw) pump laser and replaces the Lorentzian distribution of the sum frequencies we encountered in (33). The cw nature of the pump pulse implies that the arrival time of the entangled photon pair is completely undetermined. Consequently, a targeted excitation at a particular time $t$, as considered in the previous section, is impossible. Rather, within this approximation we describe a steady state

---

[9]Applying the Hölder inequality [53] to (52), and using the unitarity of the shaper functions (48), we can write the inequality

$$p_f(t) = \left| \iint \mathcal{W}_t(\omega_1, \omega_2) M_1(\omega_1) M_2(\omega_2) d\omega_1 d\omega_2 \right|^2 \leq p_f^{(s)}(t). \tag{60}$$

Because equality holds for the shaper functions satisfying (56), these must define the optimal solution.

experiment where a constant stream of entangled photon pairs gives rise to a finite population of $|f\rangle$ [54]. Our goal is to optimise this steady state population by shaping the two-photon state (62).

We consider frequency-degenerate down-converted photons, i.e. the individual photon wave packets are described by a real function $G$ symmetric around $\omega_f/2$ [55]. The two photons are therefore equally detuned from either transitions $|g\rangle \to |e\rangle$ and $|e\rangle \to |f\rangle$, since $\omega_f/2 = \omega_e + (\Delta/2)\gamma_e = (\omega_f - \omega_e) - (\Delta/2)\gamma_e$. For compactness of notation, then, we shift the frequencies as $\Omega_j = \omega_j - \omega_f/2$. The effective response function (51) for this example then reads

$$
\begin{aligned}
\mathcal{W}_t(\Omega_1, \Omega_2) &= \delta(\Omega_1 + \Omega_2)G(\Omega_1)T_t(\Omega_1 + \omega_f/2, \Omega_2 + \omega_f/2) \\
&\propto \delta(\Omega_1 + \Omega_2)G(\Omega_1)\left[\frac{1}{\Omega_1 + \left(\frac{\Delta}{2} + i\right)\gamma_e} + \frac{1}{\Omega_2 + \left(\frac{\Delta}{2} + i\right)\gamma_e}\right].
\end{aligned}
\tag{63}
$$

In this expression we dropped all prefactors, because the optimal pulse shaping operator $M$ is determined by the phase of $\mathcal{W}_t(\Omega_1, \Omega_2)$, as defined in (55,56), and therefore only[10] depends on the detuning $\Delta$ (see (17)). It is worth anticipating here that the time dependence of $T_t$ from (15) becomes a trivial phase factor $e^{-i\omega_f(t-t_0)}$, once the Dirac delta is integrated over, as we do later; for this reason, we do not spell it out explicitly in (63).

In the shifted frequencies $\Omega_j$ above, and for identical SLMs, i.e. $M_1 = M_2 = M$, Eq. (50) becomes

$$
|\psi(\Omega)|^2 M(\Omega) = \sqrt{\mathcal{N}}\,\mathcal{W}_t^*(\Omega, -\Omega)M^*(-\Omega)\int \mathcal{W}_t(\Omega', -\Omega')M(\Omega')M(-\Omega')\,d\Omega',
\tag{64}
$$

where we integrated over the variables involved in the Dirac deltas of (63). Mirroring equations (55)–(57), then, the optimal pulse shaping function reads

$$
M(\Omega) = e^{-i\mathcal{S}(\Omega, -\Omega)/2},
\tag{65}
$$

with $\mathcal{S}$ given in (55), while

$$
|\psi(\Omega)|^2 = \sqrt{\mathcal{N}}\,|\mathcal{W}_t(\Omega, -\Omega)|\int |\mathcal{W}_t(\Omega, -\Omega)|\,d\Omega.
\tag{66}
$$

These expressions presuppose that the effective response function (63) is symmetric in $\Omega$, and that $G(\Omega)$ is real, as assumed above.

In this example, the optimization ratio (61) reads

$$
E_{\text{opt}} = \frac{\left(\int |\mathcal{W}_t(\Omega, -\Omega)|\,d\Omega\right)^2}{\left|\int \mathcal{W}_t(\Omega, -\Omega)\,d\Omega\right|^2},
\tag{67}
$$

and we plot its values in Fig. 5(a) for the individual photons described by the Gaussian profile

$$
G(\Omega) = \frac{1}{(\pi\sigma^2)^{1/4}}e^{-\Omega^2/(2\sigma^2)},
\tag{68}
$$

where $\sigma$ denotes the bandwidth of the photon wave packet.

Let us remind the reader of the physical quantities we are comparing here. For the matter degrees of freedom, we want to drive the two transitions $|g\rangle \to |e\rangle$ and $|e\rangle \to |f\rangle$, of frequencies,

---

[10]The deviation $\delta$, which determines the width of the frequency-sum distribution (33), is implicitly contained in the normalization of $|\Sigma\rangle$, but for sharply correlated frequencies via the Dirac delta in (62) it does not affect the optimization.

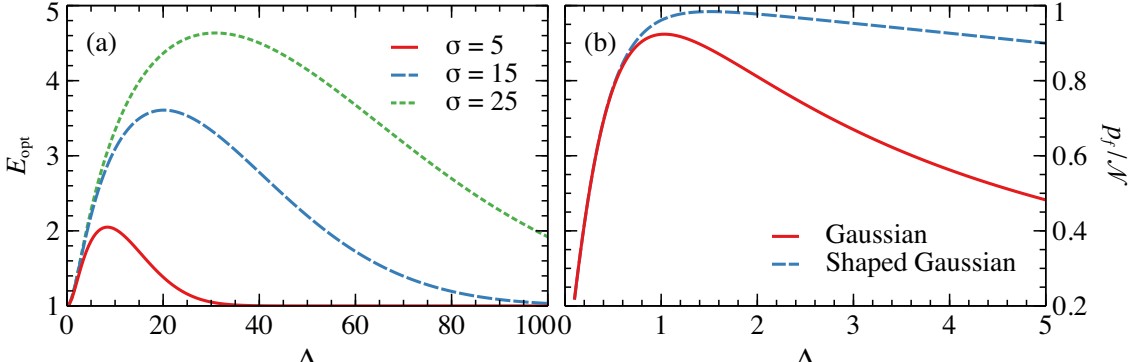

Figure 5: (a) Optimization ratio $E_{\text{opt}}$ (61) as a function of $\Delta$ (51), obtained by optimally shaping a realistic the two-photon wave function (62), given the single-photon Gaussian profile (68), for three different values of the latter's width $\sigma$. (b) Maximum achievable population, in units of $\mathcal{N}$ (31), for optimally shaped [(65), solid line] and unshaped [(62), dashed line], realistic two-photon wave functions, with Gaussian single-photon profiles (68) of width $\sigma = \Delta$, with $\Delta \geq 0.1$.

respectively, $\omega_e$ and $\omega_e + \Delta\gamma_e$, see Fig. 4(c). We excite the transitions with two photons with frequency profiles centred on $\omega_f/2 = \omega_e + (\Delta/2)\gamma_e$ and width $\sigma$. As discussed above, the two photons are always detuned by $(\Delta/2)\gamma_e$ from the electronic transitions. The probability of exciting the individual transitions depends, therefore, on the width $\sigma$ of the single-photon frequency distributions.

If $\Delta = 0$, i.e. in the case of no detuning, each photon can resonantly drive either transition $|g\rangle \rightarrow |e\rangle$ or $|e\rangle \rightarrow |f\rangle$. In this case, then, there is nothing to optimize. This can be shown formally with little algebraic manipulation: $\mathcal{W}_t(\Omega, -\Omega)$ from (63) becomes a real function, and hence carries no phase to compensate via (56). When $\sigma \ll \Delta$, instead, the photon frequencies are so narrowly distributed around $\omega_f/2$ that they are always off resonant (by $\Delta/2$) with respect to the two electronic transitions; shaping the incoming entangled wave function barely affects the excitation probability in this case. Between these two extreme cases, for fixed detuning $\Delta$, $E_{\text{opt}}$ reaches a maximum and then saturates for $\sigma \gtrsim \Delta$, when the individual photon pulses are so broad to be resonant with any transition in the matter. In Fig. 5(b) we thus consider the case where we can freely tune the bandwidth $\sigma$ of the photon pulses to match the detuning $\Delta$, and plot the maximal final state population $p_f$ achievable by both unshaped (62) and shaped (65) Gaussian wave packets. Remarkably, the latter departs rather slowly from the maximal value $\mathcal{N}$ (31) achievable by the optimal state (30). Hence, in this case, shaping can almost saturate the optimal bound $E_q$ allowed by quantum mechanics.

Finally, the remaining open question is how optimal pulse shaping (65) of the realistic entangled state $|\Sigma\rangle$ compares to the classical limit set by optimal separable pulses (37). As in Fig. 5(b), in Fig. 6 we match the width $\sigma$ of the Gaussian pulse (68) to the detuning $\Delta$, to cover both transition frequencies. As discussed for (63), the deviation $\delta$ (see (17)) does not affect $\mathcal{W}_t(\Omega, -\Omega)$; the dependence on $\delta$ visible in Fig. 6(b) and (c) stems from the optimal population achieved by the corresponding separable pulse. Our analysis finds large areas of parameter space, i.e. the entire upper right part of Fig. 6(b), for which only shaped Gaussian pulses can violate the classical limit (compare the areas in (b) and (c) which lie on the right/below the dashed line – these are the areas where an enhancement $E_q \leq 1$ with respect to the optimal separable pulse (37) is achieved).

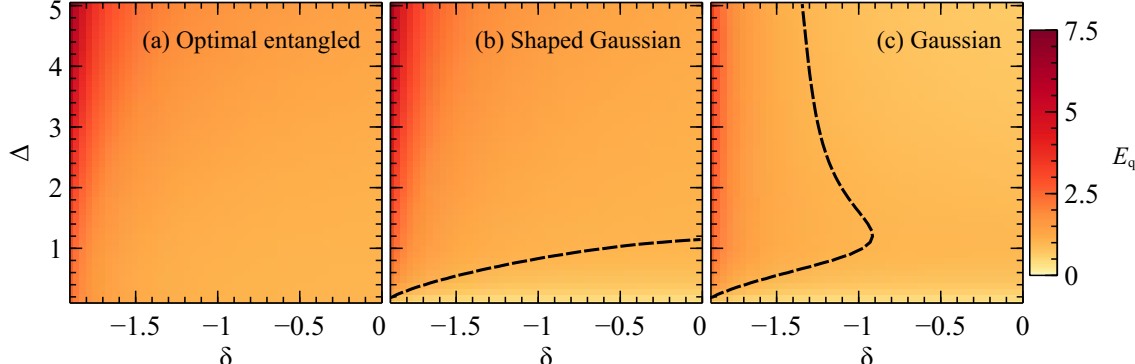

Figure 6: Comparison of the quantum enhancement $E_q$, with respect to the optimal separable pulses (37), achievable (a) via the optimal entangled two-photon wave function (32), (b) by optimal shaping (65) of a realistic pulse (62) with Gaussian single-photon distribution (68), and (c) by an unshaped realistic pulse (62), with Gaussian profile (68). In all panels we explore the ranges $\Delta \in [0.1; 5]$ and $\delta \in [-1.9; 0]$. In panels (b) and (c) the width of the Gaussian distribution is set to $\sigma = \Delta$ for each pair $(\delta, \Delta)$, to enable the comparison with the optimal separable pulses. The dashed line in (b) and (c) traces the pairs $(\delta, \Delta)$ where the realistic two-photon wave functions [shaped by (65) and unoptimized (62), respectively] yield $E_q = 1$, i.e. it demarcates the region (above/to the left of the dashed line) where the optimized states can outperform optimal separable states. Panel (a) reproduces the numerical results of [40].

## 4.4 Example II: shaping the pump

As sketched in Fig. 4(b), in this example we consider shaping a pump pulse that is subsequently down-converted. Hence, in contrast to our previous example, we will not approximate the pump laser by a spectral delta function and consider down-conversion driven by a finite-bandwidth pulse instead. The effect of pulse shaping on the entanglement of down-converted photon pairs was recently studied in [56] for a spectrally chirped pump pulse. Spectral chirp is a common effect in experiments that manifests as a quadratic phase in the pulse shape of the pump photon. Here we discuss how this phase can be counteracted via pulse shaping as described in Sec. 4.2.

We consider a two-photon state of the form

$$\Sigma(\omega_1, \omega_2) = \alpha(\omega_1 + \omega_2)\beta(\omega_1 - \omega_2). \tag{69}$$

This is an appropriate model, for instance, for photon pairs created by type-I down-conversion [57–59]. Here, $\alpha(\omega)$ is proportional to the amplitude of the pump laser driving the down-conversion, and $\beta(\omega)$ denotes the phase-matching function. To apply the optimization procedure of Sec. 4.2, we change variables to $\omega_\pm = \omega_1 \pm \omega_2$ and write the effective response function (51) as

$$\mathcal{W}_t(\omega_+, \omega_-) = \frac{1}{2}\alpha(\omega_+)\beta(\omega_-)T_t\left(\frac{\omega_+ + \omega_-}{2}, \frac{\omega_+ - \omega_-}{2}\right), \tag{70}$$

where the factor $1/2$ ensures the correct change of variables under integration. Since the pump frequency determines the sum of the down-converted photons' frequencies, we want to shape the distribution of $\omega_+ = \omega_1 + \omega_2$, which, in contrast to the previous example, is not a LOCC (local operation with classical communication, [60]) and can change the amount of entanglement in the final two-photon state. When we spectrally shape the pump pulse, we

carry out a unitary transformation on $\alpha(\omega_+)$, i.e. $\alpha(\omega_+) \rightarrow M_1(\omega_+)\alpha(\omega_+)$. Consequently, in this example we consider $M_1(\omega_+) = M(\omega_+)$ and $M_2(\omega_-) \equiv 1$, to write (50) as

$$|\psi(\omega_+)|^2 M(\omega_+) = \sqrt{\mathcal{N}}\xi^*(\omega_+) \int \xi(\omega_+) M(\omega_+) \mathrm{d}\omega_+, \tag{71}$$

with

$$\xi(\omega_+) = \int \mathcal{W}_t(\omega_+, \omega_-) \mathrm{d}\omega_-. \tag{72}$$

We can now apply our previous results: adopting the ansatz (53), and mirroring equations (55)–(57), we write $\xi(\omega_+) = |\xi(\omega_+)|e^{i\vartheta(\omega_+)}$. The optimal pulse shaping function is then

$$M(\omega_+) = e^{-i\vartheta(\omega_+)}, \tag{73}$$

with the Lagrange multiplier

$$|\psi(\omega_+)|^2 = \sqrt{\mathcal{N}}|\xi(\omega_+)| \int |\xi(\omega_+)| \mathrm{d}\omega_+. \tag{74}$$

Finally, the optimization ratio (61) becomes

$$E_{\mathrm{opt}} = \frac{\left(\int |\xi(\omega)| \mathrm{d}\omega\right)^2}{\left|\int \xi(\omega) \mathrm{d}\omega\right|^2}. \tag{75}$$

As in Sec. 4.3, to assess the usefulness of this shaping procedure we have to make $\alpha(\omega_+)$ and $\beta(\omega_-)$ in (69) explicit. Following [56], we describe the pump with a Gaussian wave packet with spectral chirp:

$$\alpha(\omega_+) = \frac{e^{-(\omega_+ - \omega_f)^2/(2\sigma^2)}}{(\pi\sigma^2)^{1/4}} e^{i\frac{\phi}{2}(\omega_+ - \omega_f)^2}, \tag{76}$$

where $\omega_f$ is the $|g\rangle \rightarrow |f\rangle$ transition frequency, $\sigma$ the pulse width, and $\phi$ the quadratic phase determining the chirp. To compare these parameters to the detuning $\Delta$ and deviation $\delta$ characterizing the matter spectrum, see (17), we first consider an "infinitely" broad phase-matching function, to avoid adding further parameters. Once this context has been analyzed, we introduce a finite width $\zeta$ also for $\beta(\omega_-)$.

**Infinitely broad phase matching** The phase-matching function describes the probability that the frequencies of the two down-converted photons differ by $\omega_- = \omega_1 - \omega_2$. In the limit of infinite phase matching, then, any detuning between the two photons is equally probable. Taking the pulse structure of Fig. 1(b) as an example, we are here considering a pulse of infinite extension along the anti-diagonal. If any difference between photon frequencies is equally probable, $\Delta$ (see (17)), the detuning between the electronic transitions $|g\rangle \rightarrow |e\rangle$ and $|e\rangle \rightarrow |f\rangle$, only determines $\omega_f$. The incoming two-photon wave function can be approximated by $\alpha(\omega_+)$, which we can take out of the integral in (72), to write

$$\xi^\infty(\omega_+) = \alpha(\omega_+)\eta^\infty(\omega_+), \tag{77}$$

where we have integrated the matter response function (15) as

$$\eta^\infty(\omega_+) = \frac{1}{2}\int T_t\left(\frac{\omega_+ + \omega_-}{2}, \frac{\omega_+ - \omega_-}{2}\right) \mathrm{d}\omega_- = \frac{-2i\sqrt{\gamma_e\gamma_f}e^{-i\omega_+ t}}{\omega_+ - \omega_f + i\gamma_f}. \tag{78}$$

In Fig. 7(a) and (b) we plot the enhancement $E_{\mathrm{opt}}$ in this limit, calculated using $\xi^\infty$ in (75). In the absence of the chirp, panel (a), we are comparing the bandwidth of the pulse $\sigma$ to

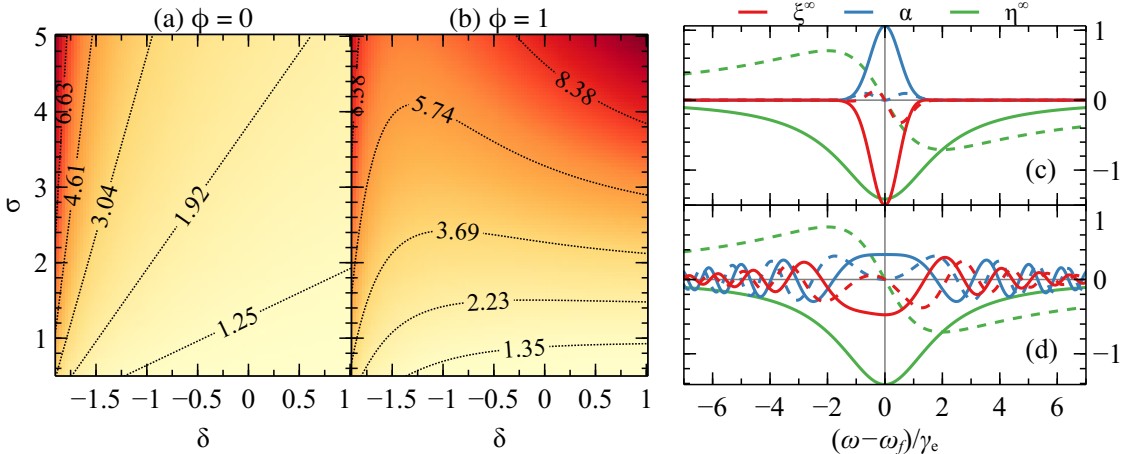

Figure 7: Left: Optimization ratio $E_{\text{opt}}$ (75), as a function of the deviation $\delta$, obtained by shaping via (73) a pump pulse with a profile (76) of bandwidth $\sigma$. Panel (a) shows $E_{\text{opt}}$ for a quadratic phase $\phi = 0$, while (b) for $\phi = 1$. Right: Comparison of the one-photon distributions $\alpha$ (76), the integrated matter response function $\eta^{\infty}$ (78), and their product $\xi^{\infty}$ (77). Here $\Delta = \delta = 0$, $\phi = 1$ and, respectively, (c) $\sigma = 0.5$ and (d) $\sigma = 5$. Continuous (dashed) lines indicate the real (imaginary) parts of the functions.

the linewidth $\gamma_f = (2 + \delta)\gamma_e$ of the two-photon transition. Following the remarks closing Sec. 4.2, the pulse shaping imposes the correct phase structure, entirely encoded in $\eta^{\infty}(\omega_+)$, to the state $|\Sigma\rangle$, maximizing its overlap with the optimal state $|\Phi\rangle$. The enhancement therefore increases when we move to negative $\delta$, i.e. when the phase of $\eta^{\infty}$ rapidly changes with $\omega_+$ within the bandwidth $\sigma$ of the wave packet $\alpha$.

The same reasoning can be applied when the chirp is present and induces a non-monotonic increase of $E_{\text{opt}}$ with $\sigma$ and $\delta$, see Fig. 7(b). For large $\sigma$, the chirp introduces significant oscillations in $\xi^{\infty}(\omega_+)$: compare panels (c) and (d). When $\gamma_f$ is also large, for $\delta > 0$ in panel (b), these oscillations also affect $\xi^{\infty}$. In this case, then, shaping via (73) adds exactly the right frequency dependence to the phase of $|\Sigma\rangle$ to counteract these oscillations.

**Gaussian phase matching**  To discuss the case of a phase-matching function with a finite width $\zeta$, we consider the Gaussian distribution

$$\beta(\omega_-) = \frac{\text{e}^{-\omega_-^2/(2\zeta^2)}}{(\pi\zeta^2)^{1/4}}. \tag{79}$$

With the definition $\chi = \omega_+ - 2(\omega_e - \text{i}\gamma_e)$ we can then write

$$\eta(\omega_+) = \int \beta(\omega_-) T_t\left(\frac{\omega_+ + \omega_-}{2}, \frac{\omega_+ - \omega_-}{2}\right) \text{d}\omega_- = 2\eta^{\infty}(\omega_+)\beta(\chi)\mathcal{G}(\text{i}\chi/\zeta), \tag{80}$$

where $\mathcal{G}$ denotes the cumulative function of the standard normal distribution [53].

In Fig. 8(b,c) we fix the quadratic phase at $\phi = 1$, and tune the parameters $\sigma$ and $\zeta$ characterizing the incoming pulse (69) via, respectively, $\alpha(\omega_+)$ and $\beta(\omega_-)$. The former determines the width of the pump pulse, and hence should be compared to the deviation $\delta$, which sets the width of the optimal two-photon wave function (32), see Fig. 1(b) and (33). The latter determines the frequency difference between the two down-converted photons, and thus should be compared to the deviation $\Delta$ between the transition frequencies $|g\rangle \to |e\rangle$ and $|e\rangle \to |f\rangle$, see Fig. 1(c) and (34).

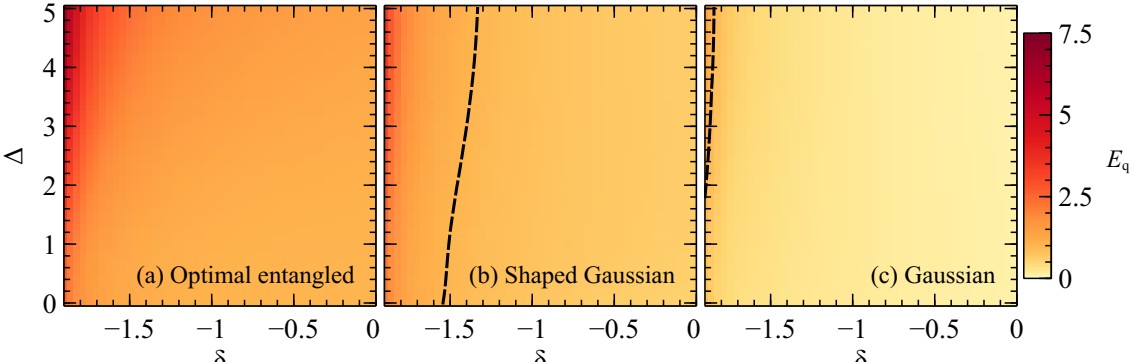

Figure 8: Comparison of the quantum enhancement $E_q$, with respect to the optimal separable pulses (37), achievable via (a) the optimal entangled two-photon wave function (15), (b) the unshaped Gaussian pulse of (69), and (c) the optimized Gaussian via the pulse shaping function of (73). For each $(\delta, \Delta)$ defining the matter response function (15), we attune the Gaussian pulse by setting its widths as $\sigma = 3\gamma_e(2+\delta)$ and $\zeta = \gamma_e(2+\Delta)$ (see text). The quadratic phase in (76) is $\phi = 1$. On the left of the dashed line the various two-photon wave functions perform better ($E_q \geq 1$) than the optimal separable pulses (37). Panel (a) reproduces the numerical results of [40].

As in Sec. 4.3, we adjust the pulse parameters $(\sigma, \zeta)$ to the matter characteristics $(\delta, \Delta)$ by setting[11] $\sigma = 3\gamma_f = 3\gamma_e(2+\delta)$ and $\zeta = \gamma_e(2+\Delta)$. This allows us to explore in Fig. 8 the same parameter space $(\delta, \Delta)$ that was considered in Sec. 3. In panel (a) we show the enhancement $E_q$ in final-state population achieved by the optimal state (32) over the classical limit given by (37), which was already presented in [40] and discussed in Sec. 3.4. In panels (b) and (c) we analogously compare the enhancement attained by, respectively, the Gaussian pulse (69) upon shaping the pump with (73), and without shaping. As shown in Sec. 3.4, the ideal two-photon state (15) maximally populates the state $|f\rangle$, and hence yields the largest $E_q$ for any fixed value of $\Delta$ and $\delta$. In panels (b) and (c) we also observe $E_q \geq 1$ in portions (on the left) of the parameter space that we demarcate with a dashed lines. There, also Gaussian pulses are able to enhance the TPA with respect to the yield of the optimal classical pulses. We recall what we concluded in Sec. 4.2: the shaped pulses can yield larger enhancement than the unoptimized pulses because the pulse shaper (73) perfectly counteracts the chirp.

## 5 Conclusions

To conclude, we analyzed optimal two-photon states to drive a two-photon transition. We first investigated the optimal state, quantified its quantum correlations via the entanglement entropy, and discussed the relation of the latter to the quantum enhancement in TPA such a state can achieve. We then drew a comparison to the optimal separable pulse, computed from the Schmidt decomposition of the response function, such that the enhancement really stems from the entanglement in the state. For the maximally achievable enhancement we also provided bounds that depend on how strongly anticorrelated the joint distribution of the frequencies is.

We then considered more realistic scenarios where a given initial two-photon state is

---

[11]To choose the specific functional dependence of $\sigma$ and $\zeta$ we initially set $\sigma = A\gamma_f$ and $\zeta = B\gamma_e(2+\Delta)$, and computed the enhancement for several choices of $A$ and $B$. This coarse numerical analysis showed that $A = 3$ and $B = 1$ are convenient values, yielding an enhancement close to the maximal, and perfectly sufficient for the remarks we intend to make in this example.

manipulated in order to enhance the two-photon transition. We defined a new optimization problem, this time for unitary operators representing local transformations of the individual photons. We derived a self-consistent equation that can be solved analytically when we consider two photons created in spontaneous parametric down-conversion. In particular, we inspected the case of spatial light modulators to shape two photons converted from a monochromatic pump laser, and then considered shaping the pump pulse directly. We found that initial two-photon states with sufficiently strong entanglement can sustain a substantial enhancement of the TPA probability, of the same order of magnitude as what is ideally achievable.

# Acknowledgements

**Funding information** This work is supported by the European Research Council under the European Union's Seventh Framework Programme (FP7/2007-2013) Grant Agreement No. 319286 Q-MAC. E.G.C. acknowledges support from the Georg H. Endress foundation. F.S. acknowledges support from the Cluster of Excellence "Advanced Imaging of Matter" of the Deutsche Forschungsgemeinschaft (DFG) - EXC 2056 - project ID 390715994.

# A   Maximization of (47)

Let us write all the terms appearing in (47) in the frequency representation, by resolving the identity in the appropriate one- or two-photon frequency space, see Sec. 2.6. If we call $\mathcal{A}[M_1, M_2] = \langle \Phi | M_1 M_2 | \Sigma \rangle$, we then have

$$\mathcal{A}[M_1, M_2] = \iint_{\mathbb{R}^2} d\omega_1 d\omega_2 \iint_{\mathbb{R}^2} d\nu_1 d\nu_2 \, \Phi^*(\omega_1, \omega_2) M_1(\nu_1, \omega_1) M_2(\nu_2, \omega_2) \Sigma(\nu_1, \nu_2). \quad (81)$$

Similarly, we define

$$\mathcal{B}_j[M_j] = \langle \psi_j | M_j^\dagger M_j | \psi_j \rangle = \iiint_{\mathbb{R}^3} d\omega_j d\nu_j d\mu_j \, \psi_j^*(\omega_j) M_j^*(\nu_j, \omega_j) M_j(\mu_j, \nu_j) \psi_j(\mu_j) \quad (82)$$

and

$$\mathcal{C}_j = \int_{\mathbb{R}} d\omega_j \psi_j^*(\omega_j) \psi_j(\omega_j). \quad (83)$$

We remind the reader that $\mathcal{B}_j$ and $\mathcal{C}_j$ are also functionals of $\psi_j(\omega_j)$ and $\psi_j^*(\omega_j)$, the wave function (and corresponding conjugate) representing $|\psi_j\rangle$. Since the latter play the role of Lagrange multipliers, however, we do not write them as arguments on the left-hand side of (82,83).

We can then rewrite (47) in the frequency representation as

$$J[M_1, M_2] = \mathcal{N} \mathcal{A}[M_1, M_2] \mathcal{A}^*[M_1^*, M_2^*] - \sum_{j=1}^{2} \{ \mathcal{B}_j[M_j] - \mathcal{C}_j \}, \quad (84)$$

where the normalization $\mathcal{C}_j$ of the $|\psi_j\rangle$ states, being a number, does not carry any dependence on the pulse shaping functions, over which we are optimizing.

To find the solution to (84), we require its functional derivatives with respect to the pulse shaping function $M_j^*$ (to obtain an expression for $M_j$) and to the Lagrange multipliers, respectively, to vanish:

$$\frac{\delta J}{\delta M_j^*} = 0, \quad (85)$$

and

$$\frac{\delta J}{\delta \psi_j^*} = 0. \tag{86}$$

Equation (86) enforces the unitarity (48), while (85) yields the general integral equation

$$\psi_j^*(\omega_j) \int_{\mathbb{R}} \mathrm{d}\mu_j \, M_j(\mu_j, \nu_j) \psi_j(\mu_j) = \mathcal{N} \mathcal{A}[M_1, M_2] \frac{\delta \mathcal{A}^*}{\delta M_j^*}(\nu_j, \omega_j), \tag{87}$$

where $j, k \in \{1, 2\}, j \neq k$, and

$$\frac{\delta \mathcal{A}^*}{\delta M_j^*}(\nu_j, \omega_j) = \iint_{\mathbb{R}^2} \mathrm{d}\omega_k \mathrm{d}\nu_k \, \Sigma^*(\omega_j, \omega_k) M_k^*(\nu_k, \omega_k) \Phi(\nu_j, \nu_k). \tag{88}$$

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
