# Peer review of "How to optimize the absorption of two entangled photons"

_SciPost Physics Core, doi:SciPost Phys. Core 4, 028 (2021)_

## Round 1 · Referee Report · Anonymous (Referee 1) · 2021-6-20

Strengths
The paper is a comprehensive account of a quantum viewpoint of
two photon absorption with emphasis on the role of entanglement.
The paper explores quantum control of the photon field to enhance
or even saturate the two photon absorption.
The authors also formulate the optimization with the use of an SLM
which is directly relevant for experiment.
two photon absorption with emphasis on the role of entanglement.
The paper explores quantum control of the photon field to enhance
or even saturate the two photon absorption.
The authors also formulate the optimization with the use of an SLM
which is directly relevant for experiment.
Weaknesses
The paper is dedicated to weak field excitation. as a result the atomic response function is almost trivial. As a result the role of interfering pathways for control is minimal.
Report
The paper is clear and well written.
The authors describe the main aspects of the theory.
The paper is therefore lengthly but considering the comprehensive
overview it is appropriate.
The authors describe the main aspects of the theory.
The paper is therefore lengthly but considering the comprehensive
overview it is appropriate.

---

## Round 1 · Referee Report · Shaul Mukamel (Referee 2) · 2021-6-23

Report
This is an interesting study since the field of quantum light spectroscopy is rapidly growing.
The authors explore the problem of what is the optimal two-photon state for entangled two-photon absorption (ETPA) for a three-level system. They showed that the optimal state is essentially the entangled photon state emitted from the model. Realistic pulse shaping and optimization relevant to current experimental design are further discussed.
The authors first present the optimal entangled-photon wavefunction for enhancing two-photon absorption in a simple three-level system and then discuss the relation between the entanglement and signal enhancement. This is certainly an interesting topic, considering the very weak entangled two-photon absorption cross section. This question has largely been answered in the paper of one of the authors titled “Theory of coherent control with quantum light” 2017 New J. Phys. 19 013009. In the first half of the manuscript, the authors mostly repeat what has been done there.
The new thing in this manuscript is the adaption of the optimization problem into a realistic experimental setting. The authors consider the optimal local pulse shaping on each photon of the pairs produced by spontaneous parametric down-conversion and the optimal shaping on the pump. However, the authors only consider a very simple and unrealistic three-level model system. In order to be experimentally relevant, the authors should address a realistic molecular model.
Overall, although this manuscript has some potentially interesting results, they are not sufficiently novel and practical enough to warrant publication in SciPost Physics.
Here are some additional comments
The langrage multiplier only guarantees that ⟨n_1 n_2 ⟩=1, this does not guarantee a two-photon state with each mode containing a single photon. This should be addressed.
The three-level model is too simple to represent a molecule. A discussion of how the complexity of a manifold of e and f states would affect the conclusions will be very useful.
Since the f-state in the model has a finite lifetime, a more meaningful target to optimize would be the fluorescence rate from the f-state instead of the f-state population.
To define quantum enhancement, the authors compare the ETPA with the signal using only the first Schmidt mode. This classical reference with a single Schmidt mode is difficult, if not impossible, to measure in experiments. A better classical reference would be the optimal unentangled state that maximize the absorption probability, which can be obtained by using the same optimization procedure as for quantum light.
5 .The critical recent work of Raymer on the (un)fesibility. of entangled TPA should be discussed. https://www.osapublishing.org/oe/fulltext.cfm?uri=oe-29-13-20022&id=451846
The authors explore the problem of what is the optimal two-photon state for entangled two-photon absorption (ETPA) for a three-level system. They showed that the optimal state is essentially the entangled photon state emitted from the model. Realistic pulse shaping and optimization relevant to current experimental design are further discussed.
The authors first present the optimal entangled-photon wavefunction for enhancing two-photon absorption in a simple three-level system and then discuss the relation between the entanglement and signal enhancement. This is certainly an interesting topic, considering the very weak entangled two-photon absorption cross section. This question has largely been answered in the paper of one of the authors titled “Theory of coherent control with quantum light” 2017 New J. Phys. 19 013009. In the first half of the manuscript, the authors mostly repeat what has been done there.
The new thing in this manuscript is the adaption of the optimization problem into a realistic experimental setting. The authors consider the optimal local pulse shaping on each photon of the pairs produced by spontaneous parametric down-conversion and the optimal shaping on the pump. However, the authors only consider a very simple and unrealistic three-level model system. In order to be experimentally relevant, the authors should address a realistic molecular model.
Overall, although this manuscript has some potentially interesting results, they are not sufficiently novel and practical enough to warrant publication in SciPost Physics.
Here are some additional comments
The langrage multiplier only guarantees that ⟨n_1 n_2 ⟩=1, this does not guarantee a two-photon state with each mode containing a single photon. This should be addressed.
The three-level model is too simple to represent a molecule. A discussion of how the complexity of a manifold of e and f states would affect the conclusions will be very useful.
Since the f-state in the model has a finite lifetime, a more meaningful target to optimize would be the fluorescence rate from the f-state instead of the f-state population.
To define quantum enhancement, the authors compare the ETPA with the signal using only the first Schmidt mode. This classical reference with a single Schmidt mode is difficult, if not impossible, to measure in experiments. A better classical reference would be the optimal unentangled state that maximize the absorption probability, which can be obtained by using the same optimization procedure as for quantum light.
5 .The critical recent work of Raymer on the (un)fesibility. of entangled TPA should be discussed. https://www.osapublishing.org/oe/fulltext.cfm?uri=oe-29-13-20022&id=451846

---

## Round 2 · Referee Report · Shaul Mukamel (Referee 2) · 2021-9-17

Report

The authors had addressed my concerns. The paper is well written and should be now accepted for publication.

---

## Round 2 · Author Response

Answer to Report 1

We thank the Referee for their report. The role of interfering pathways is indeed minimal, but rather because our system consists of three levels, not because the field is weak. The presence of multiple pathways in a more complex spectrum is discussed in J. Chem. Phys. 154, 214114 (2021), which we now cite as the new Ref. 44 in the manuscript.

Answer to Report 2

We thank Prof. Mukamel for his detailed and helpful feedback. We agree that our manuscript does not present sufficiently groundbreaking results to be published in SciPost Physics; for this reason, we did submit it to SciPost Physics Core. As Prof. Mukamel pointed out, our work “is an interesting study since the field of quantum light spectroscopy is rapidly growing.” While it’s certainly true that the foundation of this study was laid in New J. Phys. 19 013009 (2017), the first half of this manuscript does not merely repeat these earlier calculations. In particular, it * presents a more detailed and pedagogical derivation of the matter response function, starting from the case of a light-matter interaction of finite duration, and of the optimization problem; * actually quantifies the entanglement in the optimal two-photon state and relates it to the maximum enhancement it can yield, and derives limits for the possible enhancement. This analysis is entirely new. Finally, to mirror the words of Prof. Mukamel, the study adapts the optimization problem to realistic experimental settings.

All in all, we think the first half of this manuscript significantly improves and extends the ideas introduced in the NJP, while the second half applies the methods to realistic scenarios (of course starting from a theoretical point of view). The result is a self-contained package where the physics behind the optimization of two-photon absorption is clearly conveyed despite the mathematical artillery necessary in this type of problems.

(The simplicity of the three-level model we discuss in 2. below)

As for the additional comments: 1. Indeed, that we consider two-photon states only is our working assumption across the manuscript, not a consequence of the Lagrange multiplier. We have added a statement in 3.1 to make this completely transparent. 2. We agree with the Referee. The three-level model is, however, the simplest to discuss two-photon absorption, in particular in the aspects that we present in the manuscript. A more structured manifold implies an additional layer of complexity that we address in another publication: J. Chem. Phys. 154, 214114 (2021) [Ref. 44 in the manuscript]. We refer to these results in Sec. 1. 3. We have added the subsection 2.5 in the revised manuscript to discuss the relevance of our model for, e.g., fluorescence measurements. 4. We agree with these remarks, which are compatible with the message of the manuscript: our aim is to gauge the role of quantum correlations between the two photons, and to do so we have to compare the ETPA with the optimal unentangled two-photon state. In NJP (2017) [Ref. 40 in the manuscript], we showed, using the same optimization procedure as for entangled photons, that this is precisely the first pair of Schmidt modes. We fully agree that in practice these will be difficult, if not impossible, to implement. Yet they represent the best possible states and therefore, they indicate the boundary between states, whose efficiency could be matched by classical means, and those entangled states, for which this is no longer possible. We expanded Section 3.4 to highlight this point more clearly. Taking experimental difficulties into consideration in this comparison would not allow us to make general statements about the role of entanglement. 5. According to Landes et al., in the paper referred to by the referee, Ref. 33 in our manuscript, “ETPA events are not frequent enough to produce detectable signals in typical molecular systems using currently-applied SPDC sources”. Nevertheless, this conclusion is drawn “in cases in which ETPA occurs only via far-off-resonant (virtual) intermediate states”, which is not the case of our manuscript. We address this difference in Sec. 1.

---

## Round 2 · List of Changes

(All changes to the text are highlighted in red in the manuscript)

* Expanded Secs. 1, 3.1, 3.4 with additional comments.
* Added section 2.5.
* Added references 35, 36, 44, 47.
* Updated references 23 and 33.

---

## Editorial Decision

published